# Barriers and Enabling Factors Affecting Satisfaction and Safety Perception with Use of Bicycle Roads in Seoul, South Korea

**DOI:** 10.3390/ijerph16050773

**Published:** 2019-03-04

**Authors:** Bimala Sharma, Hae Kweun Nam, Wanglin Yan, Ha Yun Kim

**Affiliations:** 1Community Medicine Department, Gandaki Medical College, Pokhara, Kaski 33700, Nepal; bimalasharma@gmail.com; 2Yonsei Global Health Center, Yonsei University, Wonju 26493, Korea; 3Graduate School of Media and Governance, Keio University, Fujisawa, Kanagawa 252-0816, Japan; nsnamhan@naver.com (H.K.N.); yan@sfc.keio.ac.jp (W.Y.); 4Department of Health Administration, College of Health Sciences, Yonsei University, Wonju 26493, Korea

**Keywords:** bicycling, barriers, enabling factors, facilities, healthy cities, urban planning

## Abstract

Cycling has proven to be an important strategy in decreasing the risk of non-communicable diseases. This study aimed to discover barriers and enabling factors influencing satisfaction and safety perceptions towards the use of bicycle roads in the Seoul metropolitan area, South Korea. A cross-sectional survey of 190 youth and adult individuals was conducted. Sex, age, purpose of bicycle use, perceived safety, availability of facilities, road gradient, road width, and traffic on the bicycle road were associated with cycling regularity. Multivariate regression analysis showed that the sufficiency of bicycle parking space, moderate slope, and enough bicycle signs were significant enabling factors for satisfaction with the use of bicycle roads. Narrow bicycle roads were found to be a barrier to satisfaction with the use of bicycle roads. Moderate slope, enough bicycle signs, and enough maintenance facilities around bike roads were found to be enabling factors in the perceived safety of the use of bicycle roads, whereas traffic on the side of the bicycle road was found to be a barrier to perceived safety. Based on these findings, we conclude that healthy cities should promote cycling behavior encouraging enabling factors and initiating attempts to improve the factors that act as barriers through urban planning.

## 1. Introduction

Increasing regular physical activity is essential to improving public health. Previous research has found that active transportation is associated with reduced cardiovascular risk factor [1,2]. Cycling is an example of active transportation that has the potential to contribute to an increase in people’s physical activity levels [3]. In addition to health benefits, cycling is an environmentally sustainable mode of transportation [4]. Cycling daily can also provide significant economic benefits by substantially reducing household expenditure on transportation and providing a cost-effective method of exercise [5,6]. Planning for sustainable and healthy cities that include cycling has become increasingly important in the era of sustainable development.

A study from the Netherlands showed that natural and built environmental characteristics contribute to cycling duration, as well as the differential effect of environmental characteristics on cycling duration by municipality size [7]. Previous studies have shown that cycling behavior, including duration and frequency, differ based on people’s socio-demographic characteristics such as age, sex, and education level [8,9,10,11,12], and that cycling behavior also differs based on geographic variability [13,14].

Large variations exist in the use of bicycles between countries. Additionally, bicycle use varies between areas and municipalities within a country based on the geographical territory and built environment [7,15]. In 1995, South Korea enacted the ‘Promotion of the Use of Bicycles Act’ and has been pursuing a policy of steadily improving bicycle use [16]. In 2009, the South Korean government devised a national bike path master plan which included an infrastructure expansion scheme and covered a wide range of relevant topics such as educational programmes, publicity measures, and guidelines for building and managing bike paths. The expansion of cycling infrastructure has led to the number of bicycle users in South Korea steadily increasing, although most of them are leisure-oriented cyclists [17]. In 2016, in South Korea, the daily bicycling rate was 8.3%, and the rate of bicycling more than once a week was 25.6%. In Seoul City, the rate of daily bicycling was 9.3%, and the rate of bicycling more than once a week was 33.3%. This is higher than the overall average for South Korea [18]. According to another report, among the 23.8% of bicycle users in Seoul in 2016, 4.6% were used for transportation, and 19.2% were used for leisure [19]. To promote active transportation, Seoul City has built a total of 860.570 km of public bicycle roads with different associated facilities [19]. 

Many studies have been conducted on physical activity globally; however, little research has focused specifically on cycling. Cycling as a sustainable mode of transportation may have great potential in South Korea for dealing with non-communicable diseases. Thus, it is important to understand the environmental and socio-demographic factors that influence cycling behavior and the policy implications for the promotion of enabling environments to improve bicycling in metropolitan Seoul and other similar cities. Therefore, the current research aimed to uncover the barriers and enabling factors influencing satisfaction and safety perception with the use of bicycle roads in the northwestern part of metropolitan Seoul, South Korea.

## 2. Materials and Methods

### 2.1. Study Design and Population

A cross-sectional study was conducted in 2018; data was collected from 193 adults and youths in Seoul metropolitan, South Korea. The information was collected on 2 and 3 June 2018, using a survey. Data from three individuals were excluded due to incomplete information provided for approximately 50% of the variables at the stage of analysis, resulting in a final sample size of *n* = 190. 

### 2.2. Study Area

The study was conducted in the north-western part of the Seoul metropolitan area in three areas: Eunpyeong-gu, Mapo-gu, and Seodaemun-gu.

### 2.3. Sampling

The sample size was calculated using the G-Power program. The recommended total sample size was 177 with actual power estimated at 0.80. The information was collected from 193 individuals; and 190 participants were included in the analysis for the study.

### 2.4. Information Collection

A self-reported questionnaire was provided to the participants to be filled by themselves. The questionnaire examined the barriers and enabling factors in the Seoul metropolitan area and consisted of questions regarding socio-demographic factors, cycling behavior, satisfaction, and safety perception in the city.

### 2.5. Data Analysis

Statistical Package for Social Science (SPSS) version 24.0 (SPSS Inc., Chicago, IL, USA) was used to analyze the data. Data were summarized using descriptive statistics; and a chi-square test was used to determine the association between type of cyclist and socio-demographic and influential factors. Multivariate regression models were computed with significance level of α = 0.05 to determine which factors influenced satisfaction and perceived safety with the use of a bicycle road among cyclists. Satisfaction with the bicycle road and perceived safety were measured using a continuous scale. Out of the 17 factors, two factors were excluded from the regression analysis because VIF values were > 3.

### 2.6. Measurement of Variables

#### 2.6.1. Dependent Variable

To measure participants’ satisfaction with the use of bicycle roads, the following question was asked: ‘Are you generally satisfied with the use of bicycle roads in the northwestern part of Seoul (Eunpyeong-gu, Mapo-gu, and Seodaemun-gu)?’ Participants responded by selecting one of the following responses: very dissatisfied, dissatisfied, usual, satisfied, and very satisfied. To measure the perceived safety of using bicycle roads we asked: ‘How safe do you think it is to use a bicycle road?’ Participants responded by selecting one of the following responses: very low, low, usual, high, and very high. For both questions, the options corresponded to numbered options labelled one through five.

#### 2.6.2. Independent Variables

Socio-demographic variables: Participants were asked to report on several socio-demographic variables, including sex, age, residence, type of residence, educational level, income level, number of bicycles at home, number of cars, and occupation.Cycling behavior: To determine what type of cyclist the participant was, and to determine the average amount of cycling per month, they were asked: ‘How often did you use your bicycle during the past month?’ The type of cyclist was categorized by how often the respondent rode a bicycle during the last month. Respondents who said, ‘I rarely ride a bicycle in the last month’, ‘very occasionally’, and ‘sometimes’ were grouped as a non-cyclist/irregular cyclist. Respondents who reported ‘frequently’ or ‘very often’ were categorized as a regular cyclist. Participants were also asked: ‘For how many minutes do you normally use a bicycle?’ to measure the average time spent cycling.Barriers and enabling factors: The following questions were asked to assess what barriers and enabling factors contributed to participants cycling habits as shown in Table 1. All factors were framed with the overarching question ‘How much do the following items affect you when you use your bicycle?’ Participants responded using a 5-point Likert-type scale ranging from 1 (very little effect) to 5 (very influential effect).

### 2.7. Ethical Considerations

The Institutional Review Board (IRB) provided approval for this study (IRB: 1041849-201806-SB-053-02). Informed consent was obtained from each respondent prior to data collection. The objective of the survey and research was made clear to participants before data was collected.

## 3. Results

Of the total respondents, 73.2% were males, 18.9% were youths under the age of 18, and 8.9% were above older than 65. Of the total respondents, 34.2% reported that they were regular cyclists and the rest of the respondents cycled rarely, occasionally, or sometimes within the last one month before taking the survey. On average, participants reported cycling 9 days (±8.7 days) in the month preceding the study, with 88.7 (±82.7 min) minutes per day being the average time spent cycling.

In regard to satisfaction with the use of cycle roads in the metropolitan area, 26.3% of respondents were very dissatisfied or dissatisfied, 23.7% were satisfied, and 10.5% were very satisfied. Regarding perceptions of safety, 27.4% of respondents reported high levels of safety, and 3.2% reported very high levels of safety in using the bicycle roads. However, 7.4% and 20.5% mentioned feeling very low and low levels of safety when using the bicycle roads in the study area (Table 2).

A total of 17 items were given to the respondents to rate and report their opinions on the factor affecting bicycling riding in the Seoul metropolitan area. The factors were presented based on their mean rank (factors are rank ordered based on their means). Conflict with pedestrians on the bicycle side of the road ranked as the top concern of cyclists, followed by the level of traffic on the bicycle road (Table 3).

Both gender and age were significantly associated with the type of cyclist (i.e., regular, non-cyclist/irregular) which was based on cycling frequency over one month. However, the study area, household income level, and the number of cars available at home were not related with being an irregular or regular bicycle rider among study respondents. Regarding the various uses of cycling, among those who used it to commute, 75% used it regularly, while only 31.5% cycled regularly among those who used it for hobby and leisure time. There was a significant association between the purposes of cycling and the regularity of cycling. The number of bicycles available at home was also associated with the type of cyclist or the regularity with which individuals cycled over the period of one month (Table 4).

Level of satisfaction was not significantly associated with the type of cyclists. There was a significant association between perceived safety with bicycle road use and regularity of cycling. The regular cycling rate was 28.3% among those who mentioned they felt themselves unsafe while cycling and it was 48.3% among those who reported they felt safe while cycling. Respondents mentioned that cycling was more common in places where more related facilities such as convenience stores, restaurants, restrooms, shelters, etc. were available. This shows availability of facilities was associated with the regularity of cycling in the study area. Respondents mentioned that road slope was a critical and influential factor for cycling. A significant association was found between road gradient and regularity of the cycling. Another important factor for cycling was road width, and there was a significant association between the regularity of the cycling and road width. A lot of traffic on a bicycle road was another significant factor influencing the regularity of cycling (Table 5).

The regression analysis showed that 19% of the satisfaction with bicycle use is attributed to bicycle parking space. Another significant factor in bicycle use satisfaction is road gradient, for instance, one standard unit increase in the appropriateness of the road gradient accounted for a 24% increase in the satisfaction with bicycle use among the respondents. The current study found that 30% of bicycle use satisfaction was attributable to one standard unit increase or decrease in the sufficiency of bicycle signs. We also found that a one standard unit increase in the narrowness of the bicycle road corresponded to a 16% decrease in the satisfaction level. Additionally, one standard unit increase/decrease in the average time of bicycle use per day (in minutes) corresponded to a 23% change in the satisfaction level (Table 6).

Perceived bicycle road use safety was found to be influenced by the appropriateness of the road gradient. One standard unit increase in the appropriateness of the road gradient corresponded to a 20% change in the perception of the safety level among respondents and a one unit change in the sufficiency of the bicycle signs accounted for 25.6% change in the level of safety perception. Additionally, a one standard unit change in the availability of maintenance facilities around the bike road correlated to a 25.6% change in the safety perception among the bicycle riders, while a one unit increase or decrease in traffic on the bicycle roadside corresponded to 21% fluctuation in the perception of safety (Table 6).

## 4. Discussion

This study revealed that on average the respondents engaged in cycling 9 days a month, and the average time spent cycling in a day was 88 minutes. Sex, age, purpose of bicycle use, perceived safety, availability of facilities, road gradient, road width, and traffic on the bicycle road were all associated with regular cycling.

Sex was significantly associated with the frequency of cycling; specifically, females were less likely to cycle regularly. Similarly, a study conducted in Australia showed that men were more likely to cycle for recreation and transport than women; and men tended to cycle for longer time periods [10]. Similarly, age was also associated with the cycling frequency in the study. Among the socio-demographic variables, age and sex were the two important factors influencing cycling behavior. Encouraging women and adult population for cycling can be a way to overcome physical inactivity among them. Possession of bicycles at home was another factor which influenced the frequency of cycling among the respondents in this study. Similarly, Heesch et al. [4] reported that limited vehicle access was positively associated with utility cycling. Thus, it seems to be a better way to initiate efforts to improve access to cycles in the metropolitan area.

In this study, the main purpose reported by participants for using their bicycle was as a hobby or for leisure time. This finding is supported by Shin et al. [17] in their report on Bicycle Transport Policy in Korea. In the current study there was a significant association between the purpose of bicycle use and type of cyclist based on the frequency of cycling, specifically, among those who used their bicycles for leisure and hobby, more than two-thirds were categorized as irregular users. The efforts to enhance bicycle use for leisure, as a hobby, and for cycling to work would be one of the most important public health measures for addressing the ever-increasing burden of non-communicable diseases in South Korea [20,21,22,23].

The current research showed that the sufficiency of bicycle parking spaces was one of the significant factors affecting the satisfaction of cyclists. The study also showed that adequate bicycle signs were also a significant enabling factor influencing satisfaction among cyclists. Environmental factors such as moderate slopes were also found to be an enabling factor of bicycle road use satisfaction. A study conducted in Canada has also shown that the built environment and various spatial zones have a significant influence on healthy travel decisions [11]. The current study found that narrow bicycle roads were an important barrier to satisfaction with the use of bicycle roads. These findings are supported by a study conducted in Poland that reports the main perceived barrier to cycling was linked to a lack of good cycling infrastructure in the city [24]. A lack of bicycle-friendly design was found to be a considerable barrier to greater bicycle use in an Australian study [25]. These various findings illustrate the importance of the built environment in relation to cycling facilities and bicycle roads. To enhance metropolitan Seoul as a healthy city, city development policies and plans should consider the built environment and facilities that enables or hinder the cycling behavior of the population. In addition, it is clear that typical geographical factors such as gradients also have influence on cycling behavior. A systematic approach is recommended for urban planning to enhance health and sustainability through active transport, which promises to be a powerful strategy for improvements in population health on a permanent basis [26].

For perceived safety, moderate slopes, enough bicycle signs, and enough maintenance facilities around bike roads were found to be enabling factors. At the same time, traffic on the bicycle roadside was found to be a significant barrier factor for perceived safety with the use of bicycle roads. A study from Poland also reported that the main perceived barriers to cycling were linked to feelings of insecurity related to the behavior of drivers, and to road maintenance during the winter [20]. In addition to this, Heesch et al. [4] found that perceived environmental factors (crime, nearby transport, and recreational destinations) were associated with utility cycling (*p* < 0.05). Similarly, the perception of safety was found to hinder bicycling in many areas of Australia [25]. 

Numerous previous studies have argued that it is necessary to separate bicycle roads from pedestrian roads and vehicle roads, and that related infrastructure should consider the matter when establishing new road or redeveloping the urban area [10,27,28].

As the existing evidence supports the efforts to promote cycling as an important contributor for better population health [24], metropolitan Seoul may use cycling promotion as a strategy of population health addressing the barrier before mentioned. Cycling behavior has dual positive impacts on population health through both physical activity and eco-friendly transportation [1,2,3,4]. Based on these findings, the current study recommends improved policies and infrastructure improvements for bicycle-related facilities and transportation systems that foster feelings of safety among cyclists. Other than slopes of bicycle roads, which are determined by the geographical feature of the city, policy formulation and implementation are necessary to deal with the variables that affect the level of satisfaction among cyclists, including sufficient bicycle parking space, installed bicycle signs and other variables that affect perceptions of safety such as installed bicycle signs and sufficient maintenance facilities. 

As the study has been conducted in one metropolitan city in South Korea, this study has the limitations of being focused on that region alone. While there has been rapid innovation of the bicycle, including the e-bike, the study did not address this issue in the study. The study did not assess the types of bicycles being used. 

## 5. Conclusions

This study revealed that the average number of cycling days among respondents was 9 days based on cycling activity in the month prior to the survey, and 34.2% were categorized as regular cyclists. Sex, age, purpose of bicycle use, perceived safety, availability of facilities, road gradient, road width, and traffic on the bicycle road were all associated with the regularity of cycling among respondents. Multivariate regression analysis showed that sufficiency of bicycle parking space, moderate slopes, and enough bicycle signs were significant enabling factors, while narrow bicycle roads were perceived as a barrier to satisfaction in the use of bicycle roads. Moderate slopes, adequate bicycle signs, and enough maintenance facilities around bike roads were enabling factors, and traffic on the bicycle roadside was a barrier to the perceived safety of using the bicycle road. Based on these findings, concerned authorities should aim to maintain enabling factors while overcoming barriers to cycling and further encouraging cycling behavior in their cities. 

## Figures and Tables

**Table 1 ijerph-16-00773-t001:** Factors that affect respondents’ use of bicycles road.

SN	Influencing Factor
1	The road condition is good.
2	Bicycle parking space is sufficient.
3	The distance to the destination is appropriate.
4	The slope is moderate.
5	There are enough bicycle roads available.
6	The distinction between the bicycle road and the pedestrian walkway is appropriate.
7	The distinction between the bicycle road and the vehicle road is appropriate.
8	Bicycle roads are not continuous.
9	Road marking is sufficient.
10	Bicycle signs are sufficient.
11	There is frequent illegal parking around the bicycle roads.
12	There are plenty of amenities around the bike roads.
13	There are enough maintenance facilities around the bike roads.
14	The width of the bicycle road is narrow.
15	There is conflict with the pedestrians on the bicycle side of the road.
16	There is a lot of traffic on the bicycle roadside.
17	There are obstacles on the bicycle roadside.

**Table 2 ijerph-16-00773-t002:** Characteristics of the study population.

Characteristics		Number	Percentage/Mean (±SD)
Survey area	Eunpyeong-gu	61	32.1
	Mapo-gu	69	36.3
	Seodaemun-gu	60	31.6
Sex	Male	139	73.2
	Female	51	26.8
Age group (in years)	Under 18	36	18.9
	18–34	70	36.8
	35–54	33	17.4
	55–64	29	15.3
	≥65	17	8.9
	Missing	5	2.6
Income level (10,000 KRW) per month	Under 200	79	41.6
	200~299	35	18.4
	300~399	19	10.0
	400 and above	25	13.2
	Missing	32	16.8
Type of cyclist	Non/irregular	125	65.8
	Regular	65	34.2
Mean number of days using a bicycle		176	9.1 (±8.7)
Average minute of cycle use a day		181	88.7 (±82.7)
Satisfaction with use of bicycle roads	Very dissatisfied	13	6.8
	Dissatisfied	37	19.5
	Usual	75	39.5
	Satisfied	45	23.7
	Very satisfied	20	10.5
Safety perception of bicycle road usage	Very low	14	7.4
	Low	39	20.5
	Usual	79	41.6
	High	52	27.4
	Very high	6	3.2
Purpose of bicycle use	Commute	12	6.3
	Go to school	17	8.9
	Leisure/hobby	130	68.4
	personal work	17	8.9
	Linkage with public transportation	4	2.1
	Others	8	4.2

**Table 3 ijerph-16-00773-t003:** Factors influencing bicycle road use in Seoul metropolitan.

Rank		Influencing Factor	Mean (±SD)
1	Barriers	There is conflict with pedestrians on the bicycle side of the road.	3.64 (±0.98)
2	Barriers	There is a lot of traffic on the bicycle roadside.	3.58 (±0.97)
3	Barriers	There is frequent illegal parking around the bicycle roads.	3.56 (±1.24)
4	Environment	The distance to the destination is appropriate.	3.53 (±.944)
5	Barriers	There are obstacles on the bicycle road.	3.44 (±0.98)
6	Barriers	The width of the bicycle road is narrow.	3.33(±1.04)
7	Environment	The slope is moderate.	3.32 (±998)
8	Barriers	Bicycle roads are not continuous.	3.31 (±1.20)
9	Environment	The road condition is good.	3.29 (±1.03)
10	Environment	There are enough bicycle roads available.	2.99 (±1.32)
11	Facilities	Bicycle parking space is sufficient.	2.95 (±1.00)
12	Facilities	Road markings are sufficient.	2.92 (±1.06)
13	Environment	The distinction between bicycle road and the pedestrian walk is appropriate.	2.86 (±1.17)
14	Facilities	There are plenty of amenities around the bike roads.	2.82 (±1.06)
15	Facilities	Bicycle signs are sufficient.	2.80 (±1.10)
16	Environment	The distinction between bicycle road and vehicle road is appropriate.	2.77 (±1.21)
17	Facilities	There are enough maintenance facilities around the bike roads.	2.58 (±0.99)

**Table 4 ijerph-16-00773-t004:** Association between the type of cyclist and socio-demographic factors.

Characteristics	Non/Irregular Cycling	Regular Cycling	Chi-Square	*p* Value
**Survey area**	42 (68.9)	19 (31.1)	1.971	0.373
Eunpyeong-gu	41 (59.4)	28 (40.6)		
Mapo-gu	42 (70.0)	18 (30.0)		
Seodaemun-gu				
**Sex**				
Male	79 (56.8)	60 (43.2)	18.450	0.000
Female	46 (90.2)	5 (9.8)		
**Age group**				
Under 18	13 (36.1)	23 (63.9)	21.643	0.000
18–34	57 (81.4)	13 (18.6)		
35–54	21 (63.6)	12 (36.4)		
More than 55	30 (65.2)	16 (34.8)		
Income level (10,000 KRW) per month				
Under 200	59 (74.7)	20 (25.3)	1.11	0.775
200~299	25 (71.4)	10 (28.6)		
300~399	14 (73.7)	5 (26.3)		
400 and above	16 (64.0)	9 (36.0)		
**Number of bicycles at home**				
0	27 (87.1)	4 (12.9)		0.001
1	47 (74.6)	16 (25.4)		
2 or more	43 (53.8)	37 (46.3		
**Vehicle availability (car)**				
0	26 (70.3)	11 (29.7)	0.905	0.636
1	68 (63.0)	40 (37.0)		
2 or more	31 (68.9)	14 (31.1)		
**Purpose of bicycle use**				
Commute	3 (25.0)	9 (75.0)	11.937	0.036
Going to school	10 (58.8)	7 (41.2)		
Leisure/hobby	89 (68.5)	41 (31.5)		
Personal work	14 (82.4)	3 (17.6)		
Linkage with public transportation	3 (75.0)	1 (25.0)		
Others	5 (62.5)	3 (37.5)		

**Table 5 ijerph-16-00773-t005:** Association between the type of cyclist and influential factors for cycling.

Factors	Non/Irregular Cycling	Regular Cycling	Chi-Square	*p* Value
**Satisfaction level with bicycle use**				
Unsatisfied/very unsatisfied	33 (66.0)	17 (34.0)	2.837	0.242
Normal	54 (72.0)	21 (28.0)		
Satisfied/very satisfied	38 (58.5)	27 (41.5)		
**Safety perception with bicycle use**				
Low/very low	38 (71.7)	15 (28.3)	7.341	0.025
Usual	57 (72.2)	22 (27.8)		
High/very high	30 (51.7)	28 (48.3)		
**Availability of facilities at bicycle road**				
Very little/little effect	50 (73.5)	18 (26.5)		0.023
Usual	47 (70.1)	20 (29.9)		
Influential/very influential	27 (50.9)	26 (49.1)		
**Moderate gradient**				
Very little/little effect	28 (84.8)	5 (15.2)	7.249	0.027
Usual	49 (65.3)	26 (34.7)		
Influential/very influential	48 (58.5)	34 (41.5)		
**Road width**				
Very little/little effect	64 (71.9)	25 (28.1)	6.105	0.047
Usual	41 (68.3)	19 (31.7)		
Influential/very influential	20 (50)	20 (50)		
**Traffic on bicycle road**				
Very little/little effect	79 (73.1)	29 (26.9)	7.135	0.028
Usual	27 (51.9)	25 (48.1)		
Influential/very influential	17 (63.0)	10 (37.0)		

**Table 6 ijerph-16-00773-t006:** Regression analysis of the factors affecting satisfaction and safety perception with the use of bicycle roads in the northwestern part of Seoul.

Factors	Satisfaction with Bicycle Road Use	Safety Perception with Bicycle Road Use
	Standardized Beta	*p* Value	Standardized Beta	*p* Value
(Constant)		0.004		0.000
Good road condition	0.007	0.933	0.037	0.688
Sufficiency of bicycle parking space	0.191	0.032	0.058	0.537
Appropriateness of the distance to the destination	−0.074	0.416	−0.066	0.494
Moderate slope	0.243	0.011	0.204	0.043
Enough bicycle roads available	0.013	0.891	−0.053	0.601
An appropriate distinction between bicycle road and vehicle road	−0.151	0.088	−0.129	0.171
Not continuous bicycle roads	0.003	0.971	−0.065	0.415
Enough bicycle signs	0.305	0.001	0.256	0.007
Frequent illegal parking around bicycle roads	0.087	0.269	0.048	0.564
Plenty of amenities around bike road	0.087	0.314	−0.028	0.757
Enough maintenance facilities around bike road	−0.037	0.663	0.256	0.005
The narrowness of bicycle road	−0.163	0.049	−0.070	0.426
Conflict with pedestrians on bicycle roadside	−0.002	0.983	−0.008	0.924
Traffic on the bicycle roadside	0.000	0.998	−0.213	0.021
Obstacle on the bicycle roadside	−0.067	0.459	−0.023	0.811
Number of days spent cycling	−0.092	0.256	0.114	0.185
Average number of minutes spent cycling	0.234	0.001	−0.062	0.410
Type of cyclist	−0.038	0.641	−0.035	0.684
*R* Square	0.402		0.324	
Adjusted square	0.328		0.240	
*SE* of the estimate	0.874		0.842

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
