# Peer review of "Barriers and Enabling Factors Affecting Satisfaction and Safety Perception with Use of Bicycle Roads in Seoul, South Korea"

_ijerph, 2019, doi:10.3390/ijerph16050773_

Reviewer 1 Report

A solid and relevant quantitative research on the contribution and role of the bicycle as an individual means of transport in relation to health. Variables can be seen as design principles for the built environment. You refer to a Dutch investigation. The bicycle seems to be in crisis because of the rapid innovation of the bicycle, including e-bike. It is clear that there is a difference between geographical cultures and that this study has the limitation of the typical Korean situation.

Author Response

Dear Reviewer,                                                                                                          

We would like to express our sincere gratitude to you for your time and effort to review our manuscript; and we appreciate your valuable comments that certainly improve the quality of the manuscript. The revisions we made in the manuscript are highlighted in red color.

Reviewers’ Comments

A solid and relevant quantitative research on the contribution and role of the bicycle as an individual means of transport in relation to health. Variables can be seen as design principles for the built environment. You refer to a Dutch investigation. The bicycle seems to be in crisis because of the rapid innovation of the bicycle, including e-bike. It is clear that there is a difference between geographical cultures and that this study has the limitation of the typical Korean situation.

Reply to reviewer

We would like to sincerely acknowledge your time and efforts to review our manuscript.

1.      Yes, we agree bicycle seems in crisis because of the rapid innovation of the bicycle, including e-bike. Although we did not ask about the type of cycle they are using, we have added the points in the discussion section (line,250-253)

2.      We would like to acknowledge for your comments that this study has the limitation of the typical Korean situation. This point has been inclined in the discussion (line,249-250)

Reviewer 2 Report

the contribution is very interesting and significant as for ethical and urban policy purposes it poses.

It is well organised as clear.

The only criticality is the coherence between the contents of the analysis and the health issue, since the latter wasn't adequately developed; just some literature background was provided, but no significant evidence about the relation between cycling and public health.

Some insight could help the paper to be more widely appreciated.

Author Response

Dear Reviewer,                                                                                                          

We would like to express our sincere gratitude to you for your time and effort to review our manuscript; and we appreciate your valuable comments that certainly improve the quality of the manuscript. The revisions we made in the manuscript are highlighted in red color.

Reviewers’ comments

The contribution is very interesting and significant as for ethical and urban policy purposes it poses.

It is well organized as clear.

The only criticality is the coherence between the contents of the analysis and the health issue, since the latter wasn't adequately developed; just some literature background was provided, but no significant evidence about the relation between cycling and public health.

Some insight could help the paper to be more widely appreciated.

Reply to reviewer’s comment

Thank you so much for your critical analysis and very important comments. We have added public health implication of the study in the discussion section (line, 195-198, 206-209, 221-226, 238-241)

Reference numbers 20-23 and 26 are added
